# UNSUPERVISED LEARNING OF AUTOMOTIVE 3D CRASH SIMULATIONS USING LSTMS

## ABSTRACT

Long short-term memory (LSTM) networks allow to exhibit temporal dynamic behavior with feedback connections and seem a natural choice for learning sequences of 3D meshes. We introduce an approach for dynamic mesh representations as used for numerical simulations of car crashes. To bypass the complication of using 3D meshes, we transform the surface mesh sequences into spectral descriptors that efficiently encode the shape. A two branch LSTM based network architecture is chosen to learn the representations and dynamics of the crash during the simulation. The architecture is based on unsupervised video prediction by an LSTM without any convolutional layer. It uses an encoder LSTM to map an input sequence into a fixed length vector representation. On this representation one decoder LSTM performs the reconstruction of the input sequence, while the other decoder LSTM predicts the future behavior by receiving initial steps of the sequence as seed. The spatio-temporal error behavior of the model is analysed to study how well the model can extrapolate the learned spectral descriptors into the future, that is, how well it has learned to represent the underlying dynamical structural mechanics. Considering that only a few training examples are available, which is the typical case for numerical simulations, the network performs very well.

## 1 INTRODUCTION

Data driven virtual product design is nowadays an essential tool in the automotive industry saving time and resources during the development process. For a new car model, numerical crash simulations are performed where design parameters are changed to study their effects on physical and functional properties of the car such as firewall intrusion, weight, or cost (Fang et al., 2017). Since one simulation run takes a couple of hours on a compute cluster, running a large number of simulation is not feasible. Therefore, a system that is able to use a limited dataset and predict new simulations would make the development process faster and more efficient.

The rise of deep neural networks (DNNs) in recent years encourages further research and industrial usages. Besides manifold research for autonomous driving, it is natural for the automotive industry to seek and evaluate the possible applications of DNNs also in the product design stages. As an example, we investigate car crash tests, in which for example the plate thickness of certain parts strongly influences the bending behavior of structural beams and as a result also the intrusion of the firewall into the passenger compartment. Here, numerical crash simulations for different variations of such thicknesses are used as a dataset for learning. The aim is to design a system based on a DNN architecture that learns the crash behavior and would be able to imitate the crash dynamics.

Car crash simulations are based on a mathematical model of the plastic deformations and other physical and mechanical effects. They are defined on a computing mesh of currently up to three million points and up to a hundred time steps are stored. Each data instance is a simulation run—of pre-selected parts and/or time steps—that is very high dimensional. Working with this data directly exasperates any machine learning (ML) method, but a transformation of this data presented in Iza-Teran & Garcke (2019) allows to obtain a new representation that uses only a small number of coefficients to represent the high resolution numerical solutions. The transformed representation is employed here to compress the mesh geometries to feature sets suitable for neural networks, while avoiding to directly handle geometries in the machine learning method. This way, a network

designed for video prediction and embedding based on a long short-term memory (LSTM) based architecture (Srivastava et al., 2015) can be adapted for mesh data. Since LSTM is a recurrent neural network that allows to exhibit temporal dynamic behavior with feedback connections, it is a natural choice for learning the 3D sequences. The aim is that the network learns the observed crash behavior including translation, rotation, or deformation of the parts in the model.

Since the contribution of this paper is using DNNs for analyzing car crash data, the related works are categorized into a group of publications in which DNNs are extended for 3D graphics and one that concerns the use of ML techniques for analyzing car crash simulations. For the latter, one typically uses different embedding techniques to obtain a low dimensional representation for the intrinsic underlying data space and to cluster simulations with similar characteristics together (Bohn et al., 2013; Diez, 2018; Garcke & Iza-Teran, 2015; Iza-Teran & Garcke, 2019; Le Guennec et al., 2018).

The majority of publications about 3D DNN tried to extend CNN for 3D space and focus on description learning and shape correspondence, also known as geometric deep learning, (Bronstein et al., 2017; Masci et al., 2015; Monti et al., 2017; Boscaini et al., 2015; 2016; Litany et al., 2017; Halimi et al., 2018; Maturana & Scherer, 2015; Su et al., 2015; Wang et al., 2017) and some developed CNN filters for unorganized point clouds (Qi et al., 2017a;b). The very active research is so far very compute resource consuming and there is no extension of ConvLSTM for 3D space to our knowledge, but for prediction one would need an LSTM (or GAN) approach.

However, a couple of very recent works introduce new feature sets and architectures for mesh embedding using autoencoders and LSTM (Tan et al., 2018b; Qiao et al., 2018; Tan et al., 2018a). The feature representation is using local shape deformations obtained by solving an optimization problem at each node and a global optimization for compensating for rotations. They have shown that after training the network, a sequences of 3D shapes as an animation can be generated by doing operations in the latent space. The bidirectional LSTM architecture is shown to outperform autoeconders (Tan et al., 2018a). An LSTM based learning network has also been proposed in Qiao et al. (2018), where the obtained feature representation is then taken as the temporal data to be feed into a CNN that takes the features and represents them in a lower dimensional latent space. This information is subsequently feed into the LSTM module.

## 2 MESH DATA AND ITS REPRESENTATION

Data collection is a bottleneck for deep learning applications. If the training data is not diverse enough, the network would neither be able to properly learn the intrinsic data space nor be able to return reasonable output for before unseen data. We focus on surface mesh data from numerical simulations of car crashes, where in the industrial setting the car model is divided into several physical car parts.

As a simple car model we use a Chevrolet C2500 pick-up truck, a model with around 60,000 nodes from the National Crash Analysis Center[1]. The data stems from numerical crash simulation[2] of a frontal crash for random variations of the plate thickness of nine structural components, a setup similar to Bohn et al. (2013). The thickness variations result in different deformation behavior. Figure 1 shows a snapshot of the crash simulation for the truck model.

The geometries of the car model and parts are available in a regular mesh format and the correspondence of vertices between different simulations and over the time of the simulation are known by their node id. Therefore, instead of working with the meshes one can simply work with vertices and treat them like organized point clouds, while being able to recover the mesh and connectivity at any time. Now, the features for training the network are obtained from these point clouds and the network outputs a feature vector that is later post-processed to a point cloud or mesh.

Instead of working directly with 3D surface meshes, a set of features is extracted for training the network. Such a feature set should be able to represent the dynamics of the crash efficiently. In Iza-Teran & Garcke (2019) a compact representation for deforming shapes has been presented for the car crash case. The approach is based on the property that the Laplace Beltrami Operator (LBO) on a surface is invariant to isometric deformations. That is, the LBO is the same if a surface mesh is de-

---

[1]from NCAC `http://web.archive.org/web/*/www.ncac.gwu.edu/vml/models.html`
[2]computed with LS-DYNA `http://www.lstc.com/products/ls-dyna`

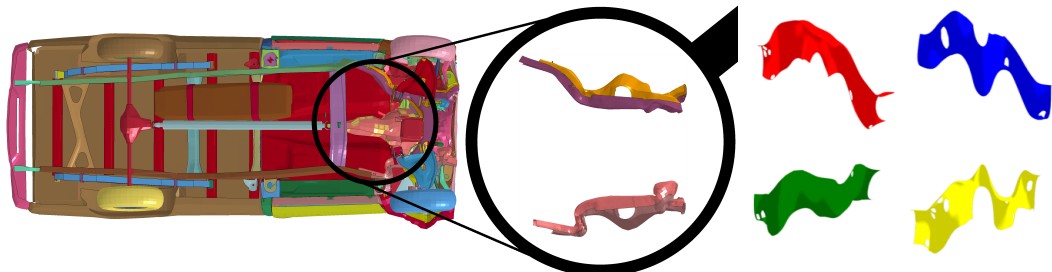

Figure 1: A snapshot of the crash simulation of the 3D model used for data collection with a zoom unto the studied longitudinal beams, from Bohn et al. (2013) (left). The four selected parts (beams) after the crash for one selected simulation, later used for illustration (right). The colors of the parts are used to display error behavior per part.

formed in such a way that it is neither stretched nor teared apart, which is the case for the considered simulations, which additionally all start from the same geometry. Consequently the eigenvectors, which form an orthogonal basis, do not change under an isometric transformation and can be used to represent mesh functions such as the deformations (as three functions, one each for $x, y, z$). The representation is obtained by projecting mesh functions onto the common orthogonal basis to compute so-called spectral coefficients. It turns out that most of the variations of the deformed shapes are concentrated in a small number of coefficients. Therefore an efficient representation is obtained using few spectral coefficients. As shown by Brezis & Gómez-Castro (2017) using suitable assumptions, for the $L_2$-approximation of functions controlled in the $H^1$-Sobolevnorm the orthonormal basis stemming from the Laplace operator provides an optimal approximation in a certain sense; this result can be extended to the LBO and functions in the Sobolev space $H^{2,2}$. Furthermore, the spectral representation can be understood as a mesh surface analogy to the Fourier decomposition of signals. This representation is introduced here for training LSTMs and allows to bypass the complexity of dealing with large meshes directly.

To formalise, in $B_{n \times m}$ we collect the first $m$ unit eigenvectors of the LBO, where $n$ is the number of points in the considered 3D shape. The spectral coefficients $C_{3 \times m}$ at one time step are obtained by

$$C_{3 \times m} = R_{3 \times n} B_{n \times m}, \tag{1}$$

where $R_{3 \times n}$ contains the $x, y, z$ coordinates of the $n$ points from the considered 3D shape. Four $B$ matrices are required for four distinct parts in the dataset, which are four distinct geometries.

Recovering the 3D shapes from their spectral coefficients is possible by

$$R'_{3 \times n} = C_{3 \times m} B^T_{n \times m}, \tag{2}$$

since $B_{n \times m}$ is orthonormal. $R'$ is an approximation of $R$ and by choosing larger $m$, the approximation error gets smaller. In other words, the more eigenvectors are used in $B_{n \times m}$, the more details are saved in the spectral representation. Figure 2 visualizes the localization and histogram of average error for reconstructing four selected parts from their spectral coefficients. The averaging is done over $205 \times 10$ samples for each part independently. Note that the bounding box of the entire data is $[2750, 4500] \times [-600, 600] \times [250, 650]$, determined over simulation time. Comparing the histogram with the tight bounding box's dimensions shows that the reconstruction error is not very high for $m = 40$, while the part with blue color coding seems to have more overall error. Note that the error is localized mostly toward the front of the parts, this area goes through very large deformations during the crash. Note that with $m = 100$ the observed maximal observed error would go down from 40 to 20.

## 3    MODEL DESCRIPTION

For frame prediction in video and to obtain latent representations a number of quite interesting applications in robotics and computer vision have been developed. Different DNN architectures have been investigated very extensively. Srivastava et al. (2015) was one of the first proposals of an

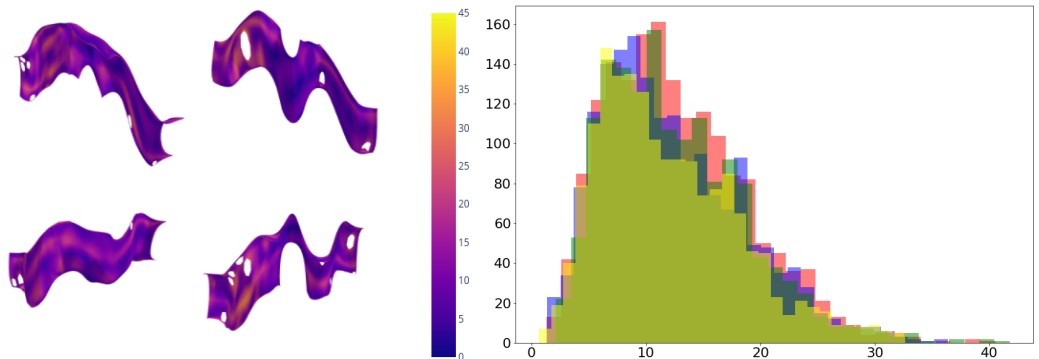

Figure 2: Localization and histogram of errors between original parts and their reconstruction from their spectral representation with $m = 40$ spectral coefficients. Histogram color coding follows the color convention in Figure 1, right.

unsupervised LSTM based representation learning method for video. The logic behind the choice is that the same operation must be applied at each step to propagate dynamics to the next step. This implies that the underlying dynamics of the dataset remains the same, i.e. the same dynamics acting on any state, at anytime, should produce the next state (Srivastava et al., 2015). Others extended this work by adding convolutional operations in LSTM, introducing new recurrent networks, or using model based architectures (Xingjian et al., 2015; Finn et al., 2016; Kalchbrenner et al., 2017; Lotter et al., 2016; Sharma et al., 2015).

Car crash simulation data can basically be considered a sequence of 3D geometries. In analogy to the video processing case, one can see each geometry in the sequence as a video frame. Inspired from the work on video prediction, we use a similar architecture for the case of car crash representation learning. We choose a two branch LSTM architecture with reconstruction and prediction decoders from Srivastava et al. (2015). The encoder LSTM maps an input sequence of $k$ time steps into a fixed length vector representation. This representation is decoded using a decoder LSTM to perform the reconstruction of the input sequence of size $k$. Receiving a few steps $l, l < k$, from the beginning of the sequence, the other decoder LSTM predicts future sequences of length $k - l$, see Figure 3, where we here use $l = k/2$ for simplicity. The architecture is in Srivastava et al. (2015) shown to have a better performance for predicting future frames compared to other approaches.

The design of the prediction encoder-decoder LSTM is same as that of the autoencoder LSTM, except that it is like a supervised method in which the decoder LSTM predicts future behavior that comes after the input sequence while the encoders hidden state will capture information about the representation of the input sequence. In comparing to an autoencoder LSTM that receives the entire $k$ time steps of the simulations and reconstruct them again, the prediction encoder-decoder LSTM receives just the first $l$ steps and predicts the remaining time steps. Therefore, the input sequence acts as seed points for generating the rest of the sequence.

This compositional architecture is supposed to address the shortcomings that one has to confront by the use of an autoencoder or encoder-predictor alone. Namely, on the one hand an autoencoder LSTM suffers from a bias by memorizing the inputs, which is not sufficient for predicting future frames. On the other hand, an encoder-decoder LSTM suffers from the bias to save information mostly about the last few frames since these carry more information for predicting the future frames, but then the representation at the end of the encoder will ignore large parts of the input. In the compositional architecture, the model is trained to also predict all of the input sequence, therefore it cannot just store information about the last few frames (Srivastava et al., 2015).

It is worth to mention that the dataset contains the translation, rotation, and deformation of car parts during the crash. Therefore the network would learn the entire degrees of freedom of the crash dynamics and would be able to imitate the crash by the prediction decoder after receiving the initial time steps as seeds.

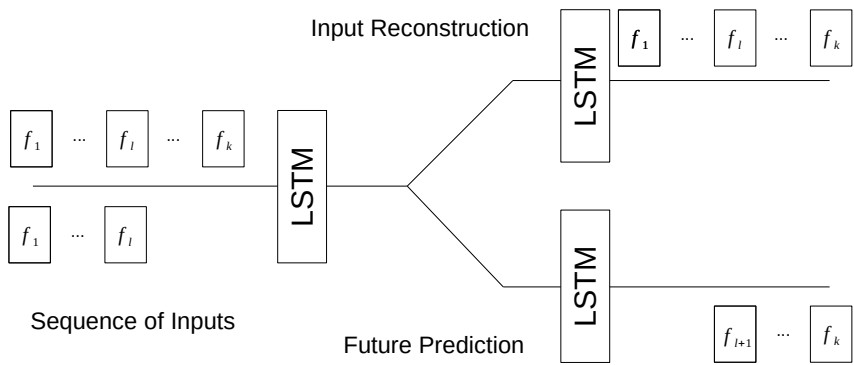

Figure 3: The two branches LSTM autoencoder network architecture. The data sequence of $f$s are feed into the network in two lengths and a joint encoder is learned. The top decoder is learning the reconstruction, while the bottom decoder is performing future prediction.

## 4 EXPERIMENTS AND EVALUATION

Although the 3D truck model consists of several parts, only the left and right structural beam, which are made of two parts each, are considered for evaluating our approach. These beams are structurally very important parts of the car and typically investigated by engineers, therefore it is justified to concentrate on these parts.

Overall 205 different simulations were performed. For each simulation, ten snapshots equally distributed in time are selected and the deformation of the two beams are extracted from it. Therefore a data sample consists of the four selected parts during ten time steps. Together, the dataset after the extraction from the crash simulations contains 205 samples and each full sample is a long concatenated vector of $4 \times 3 \times 40 \times 10$ elements. The entire dataset is divided into training and test set of 105 and 100 samples, respectively. Each data point, i.e. each time step of a simulation, is normalized by the $l_2$ norm of the features at $t = 0$ beforehand.

Note that one could consider a different scenario in which each part, out of the four, over all ten time steps is a data sample, that is treat the for beams with individual machine learning models. Since the dynamics of the two beams are coupled during the crash, considering them together can reduce the error for learning the crash dynamics by the network and we therefore consider this setup.

We employ the two branch LSTM from Figure 3, where the implementation is done with Keras. In our experiments, the encoder component of the network has 1000, the decoder has 1500, and the prediction part has 2000 LSTM units without any convolutional layer, as defining a convolutional layer for 3D graphics is not trivial and existing approaches are resource intensive. Further, we use ReLU as the activation function. Using ADAM with default parameters the entire network is trained together, which includes the encoder, reconstruction, and prediction parts over 100 iterations, until the mean squared error (over all parts) reaches the order of $10^{-7}$ and becomes stable. The training phase takes about 30 minutes on an Intel i7-7700 CPU@3.60GHz $\times$ 8. The achieved minimum during training is a trade-off between reconstruction and prediction loss functions. The prediction part of the network receives the first five time steps as inputs and generates the next five steps of the crash (which are both a vector of $4 \times 3 \times 40 \times 5$ elements).

For each grid point $j$ we compute in the following the least squares error at time $t$ and averaged over the simulations, as well as accumulated over time:

$$E^j(t) = \frac{1}{s} \sum_{i=1}^{s} \|S_j^i(t) - \hat{S}_j^i(t)\|_2, \quad AE^j = \sum_{i=1}^{k} E^j(t)$$

where $s$ is the number of simulations and $S_j^i(t), \hat{S}_j^i(t)$ are the three-dimensional (one for each direction) original mesh function and its reconstruction (or prediction) for simulation $i$ at time $t$.

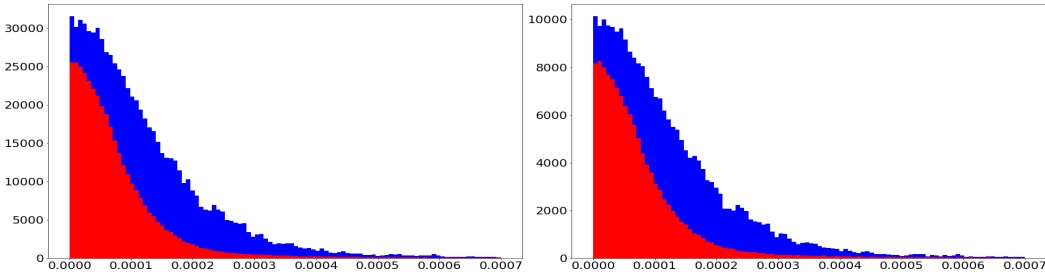

Figure 4: Histogram of the reconstruction (blue) and prediction (red) errors for the spectral coefficients by the network, left for the training data, on the right for the testing data.

### 4.1 NETWORK PERFORMANCE AND EXISTING BIFURCATION

Figure 4 shows the histograms of the reconstruction and prediction error for the training and test datasets of all four parts, only here for the spectral coefficients. The reconstruction error is larger than the prediction error since the reconstruction part recovers the entire $k = 10$ time steps while the prediction part just generates the last $l = 5$ time steps.

Bohn et al. (2013) and Iza-Teran & Garcke (2019) have reported a bifurcation for this car crash dataset. That is, two different bending behavior for the beams arise in the simulation data, which is due to changes in the plate thicknesses. One can cluster the 205 simulations into two groups of 71 and 134 for the two bending behaviors. It gives an unique opportunity to investigate the performance of the proposed system by analyzing and visualizing the error for each branch of bifurcation individually. Moreover, it is possible to see if and how well the network can recognize this bifurcation and how the spectral coefficients can preserve this.

The 105 simulations used for training include samples from both branches of the bifurcation, as do the remaining 100 for testing. The distances between ground truth and the outputs of two decoders, after decompression and re-normalization from the spectral coefficient back to 3D shapes, are visualized on a template to present the spatial localization of the average error of each bifurcation branch and its histogram is also shown for a better comparison (all for the testing dataset). Figure 5 gives the accumulated error over time. Overall, the reconstruction and prediction errors are small in relation to the bounding box of the data, the network based on the spectral coefficients is able to learn the complex structural mechanics.

It can be seen in first two histograms of Figure 5 that the errors accumulation for the reconstruction in the two branches is similar, which can also be seen in the first and second row in Figure 5. The part color coded red shows somewhat higher error values, while the blue part has more mid-range errors. The third and fourth row in Figure 5 for the prediction have very different error localization.

Considering the localization and histograms of the average error over time for the reconstruction branch of the network, one can observe that the error behavior stays roughly the same over time, i.e. the localization of error and histograms show that the error stay the same.

Regarding the localization and histogram of the average error over time for the prediction branch of the network, now the error increases in time and the behavior changes at the eighth time step. Here, one can observe that the localization of the error changes and the histograms show that the error increases more strongly for the first branch, see Figure 7 in the Appendix.

### 4.2 EMBEDDING

The encoder LSTM weights can be used for visualizing the intrinsic underlying space of the dataset. Therefore, in order to see how well the encoder learned the representation and dynamics of the car crash simulations, the weights are visualized using different markers for the two branches of bifurcation. Due to the encoder having 1000 layers, to which each sample is mapped, one needs to use some visualization techniques for embedding from higher dimensions to a lower dimensional space. We use t-SNE (Van Der Maaten & Hinton, 2008) and show the result in Figure 6. As can be seen, the bifurcation is shown as two well-separated clusters. There are a few points from each

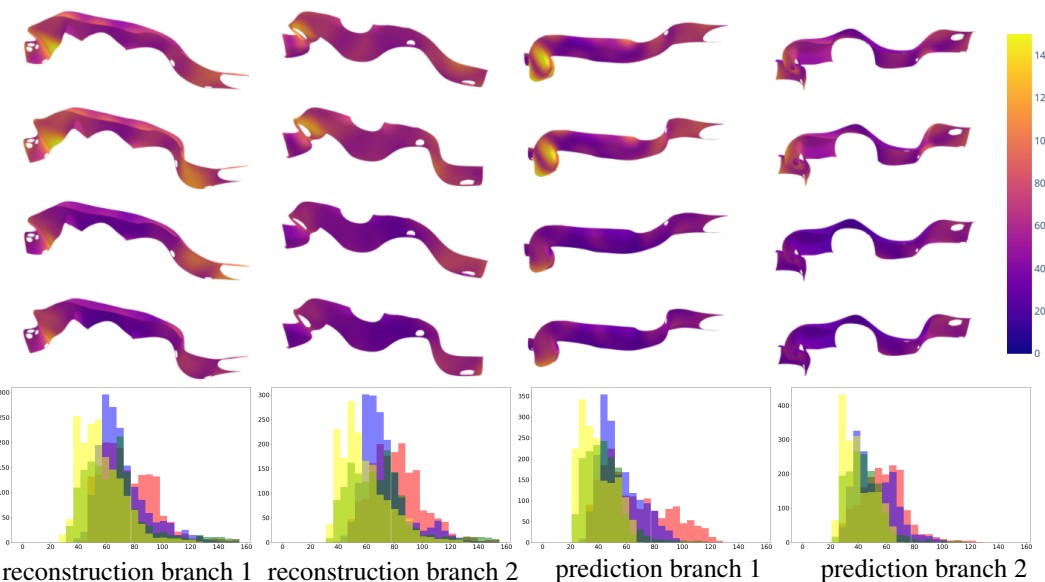

reconstruction branch 1   reconstruction branch 2   prediction branch 1   prediction branch 2

Figure 5: Localization and histogram of the error accumulation AE. First and third rows show the reconstruction and prediction error, respectively, for the first bifurcation branch. Second and forth rows show the reconstruction and prediction error, respectively, for the second bifurcation branch. The error is shown on a representative final time step of each branch to illustrate the different deformation behaviors. Histogram color coding follows the color convention in Figure 1, right.

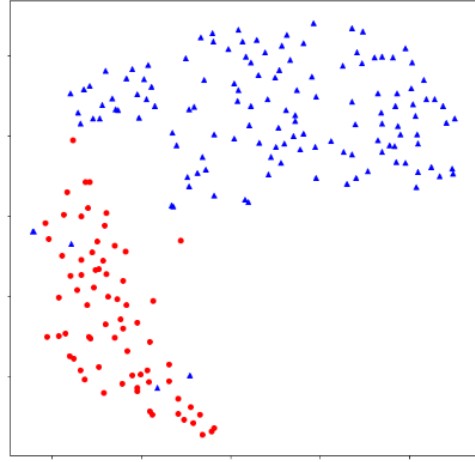

Figure 6: 2D embedding of LSTM's reconstruction weights. Two branches of the bifurcation are well separated in the 2D visualization made by t-SNE. First bifurcation branch is marrked by red dots and the second one by blue triangles.

bifurcation branch that are misaligned with the rest of their branch members. This might be because of using the spectral coefficients as an approximation which leads to losing some details about the bifurcation, therefore increasing $m$ might rectify the issue. Or a 2D embedding is not sufficient for properly visualization the encoder LSTM weights. Nevertheless, this 2D visualization proves that the network was able to learn the complex dynamics of the crash since the bifurcation is well represented by the encoder weights.

# 5 Conclusions and future work

Video frames prediction has been in the center of attention of researchers for a while, but there has been only very few extensions of these works to the 3D case so far. The problem is addressed here by introducing spectral coefficients to encode functions on the geometry together with a two branch LSTM based architecture without any convolutional layer, which has already proven to be feasible for video embedding and future frames prediction. The employed LBO basis and the resulting spectral coefficients provide a trade-off between accuracy and required computational resources. We encode the 3D shapes by a set of features using the eigenvectors of the LBO.

For empirical evaluation, a dataset is employed from a set of numerical simulations of a car during crash under different design conditions, i.e. plate thickness variations. The appearance of a bifurcation during the crash in the dataset, motivates an error analysis done for both groups to see how good the network performs in the presence of a bifurcation. In both branches, the network is able to perform very good predictions, while we observe different error localisations for reconstruction versus prediction. Moreover, the 2D visualization of the reconstruction branch shows the bifurcation as two clusters. In any case, from a relatively small number of data, the proposed network using spectral coefficients is able to learn complex dynamical structural mechanical behaviors.

Future work could go toward scaling the pipeline for learning the crash dynamics of the entire car and larger mesh sizes, which increases the needed computational effort. On the other hand, one might be able to use smaller number of eigenvectors by not simply selecting the first few ones, but those with a large variance in the spectral coefficients of the data set. Furthermore, in practical settings, re-meshing of the parts can take place, here using spectral coefficients can ease this step since one can encode shapes with different vertices number to fixed size feature vectors, as long as the geometry is (approximately) isometric. Still, there is the overall question, if and how a trained network can be evaluated for changed geometries (relevant question for any 3D DNN approach introduced so far) or different crash setups. Moreover, adding design parameters could also improve the accuracy but requires modifications of the networks architecture.

For practical applications, as each crash simulation requires hours of heavy computation running computational solvers on a large cluster, a system that is able to learn the representation of experiments with very few training data and generate the predicted simulation results for new design parameters would save much resources. Moreover, the ultimate goal of research along this direction would be a data driven system that receives very little information about the simulation (like design parameters) and output the crash sequences with minimum error.

Another application of the current system could be feasibility detectors while running the simulation on the compute cluster. Using the network, one could check if the simulation goes well or if for some reasons it should be terminated. From the current stage of the system, one would be able to generate the parts of the future simulation simply by extrapolating the learned spectral coefficients from a few initial time steps, which are already computed on the cluster, as inputs. If the distance between network predicts and simulation gets very large over the iterations, the simulation can be terminated since it failed the feasibility check.

Further, related works such as Qiao et al. (2018) introduce a specific feature set and LSTM autoencoders, where also graph convolution operation is required. This approach could be applied for car crash data under the assumption that the local optimization can still be applied for large deformations as the ones occurring in our applications. Further, the resulting features are long vectors, which results in 8 hours for learning on a CPU/GPU system for a data set similar in size to ours, where we need 30 minutes. Nevertheless, a comparison of these two approach will be worthwhile future work.

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

# A   APPENDIX

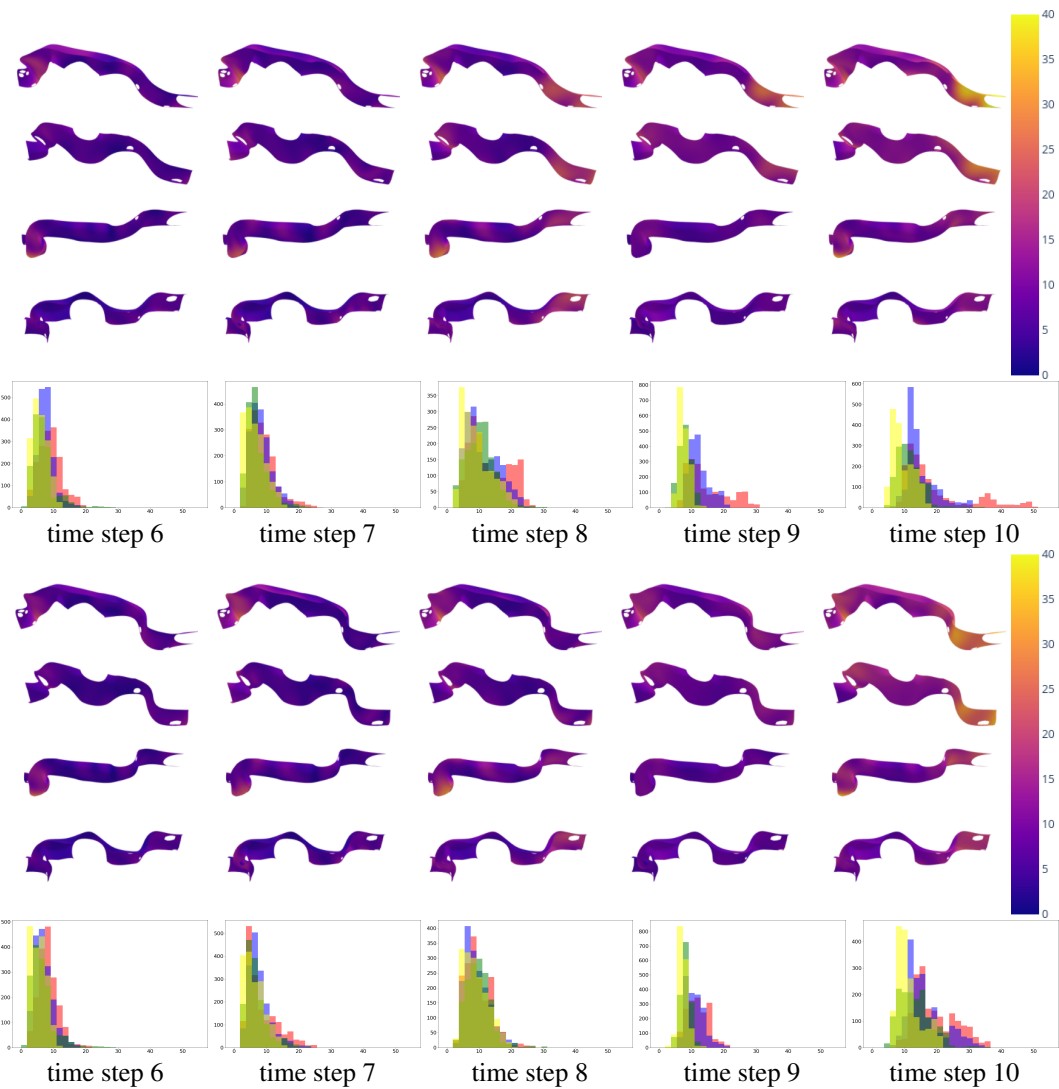

Figure 7: Localization and histogram of the error accumulation AE for the prediction per time step. Top rows show the prediction error for the first bifurcation branch. Bottom rows show the prediction error for the second bifurcation branch. The error is shown on a representative final time step of each branch. Histogram color coding follows the color convention in Figure 1, right.

