# OpenReview forum: "Unsupervised Learning of Automotive 3D Crash Simulations using LSTMs"
_ICLR.cc/2020/Conference — Reject_

### Official Review · AnonReviewer3 · 2019-10-23
**Official Blind Review #3**

**Rating:** 3

**Review:**

Summary.
Long short-term memory (LSTM) networks are trained to learn the underlying dynamics of 3D structures -- An encoder LSTM encodes a sequence of 3D mesh representations and two decoder LSTMs reconstruct the input itself and predict the future structural geometry. Such structural deformation sequences are collected from a simulation and used to train and evaluate the model.

Strengths.
1. Deep neural network architecture is applied to learn a 3D mesh structure deformation. Its approach looks reasonable to me -- (i) a high-level (user-specified) feature is extracted to represent 3D structural mesh, and (ii) LSTMs are trained in this feature space.

2. The paper shows that LSTM can learn the underlying dynamics of 3D structural deformation (but since it does not provide any comparison with other work, I cannot determine the prediction error is within a reasonable bound for this task).

Weaknesses.
1. Weak technical and theoretical novelty.
An existing LSTM-based seq-to-seq style architecture is simply chosen and applied to the 3D mesh reconstruction and prediction task. LSTM networks have been successfully applied to various tasks, thus it would be less impactful even if a paper shows LSTM works well in the specific task.

2. Missing comparison with existing work.
(As far as I know) the task of predicting 3D structure deformation has long been studied and various simulation and analysis tools have widely used in academia and industries. The paper, however, does not thoroughly compare or cite any prior work in this area. As a reader from a different area, I would like to see more thorough analysis to support (i) what are the main bottlenecks of the conventional approaches, (ii) how the proposed data-driven approach can help to address these issues, and (iii) how the proposed approach can further extend.

For example, the paper mentions (as the main reason to use the deep neural network) that “Since one simulation run takes a couple of hours on a compute cluster, running a large number of simulation is not feasible” (in the 1st paragraph of Introduction). Can the authors provide an analysis of flops (floating-point operations per second)?

3. Simulated data for training.
205 samples are collected from a simulation to train and validate the proposed model and half an hour has taken to train this model. Judging from the common practices, does this imply the underlying dynamics function (of a simulation) is trivial to learn?

Also, only two parts (i.e. the left and right structural beam) of the whole 3D structure are analyzed. This implies that relations between any other parts are not considered and only a sole part is individually analyzed.

Minor concerns.
- Typo (In Figure 1 caption, ‘undo’).
- “Due to the encoder having 1000 layers”?
- Labels are needed in Figures.

**Experience Assessment:**

I do not know much about this area.

**Review Assessment: Checking Correctness Of Derivations And Theory:**

I assessed the sensibility of the derivations and theory.

**Review Assessment: Checking Correctness Of Experiments:**

I assessed the sensibility of the experiments.

**Review Assessment: Thoroughness In Paper Reading:**

I read the paper at least twice and used my best judgement in assessing the paper.

---

### Official Review · AnonReviewer1 · 2019-10-24
**Official Blind Review #1**

**Rating:** 3

**Review:**

The paper presents an LSTM model to predict the evolution for 3D models for crash tests. The core model is a video prediction model (Srivastava et al. 2015). Instead of using the original 3D geometry, the authors propose to predict a feature representation based on prior work (Brezis & Gomez-Castro 2017, Bohn et al. 2013, Teran & Garcke 2019).

Strength:
 + Interesting problem
 + Good writing
Areas of improvement:
 - Technical contribution
 - Experiments

The paper is quite easy to read, although it would help to clearly separate prior work from the actual technical contributions. The problem of simulating 3D meshes instead of raw pixels is interesting.

My main issue with the paper was that I had a hard time seeing a contribution. It is unclear what is technically new, or if everything was just copied from prior work. It was also unclear which parts of the technical approach mattered? Is the LSTM the right solution, or are there other simple baselines to compare to?

Finally, the experiments are not very intuitive. It is unclear what impact the presented technique has on the down-stream tasks of crash testing. Does it improve the current simulation of crash testing? Is the prediction error acceptable to replace certain safety tests?

**Experience Assessment:**

I have read many papers in this area.

**Review Assessment: Checking Correctness Of Derivations And Theory:**

I assessed the sensibility of the derivations and theory.

**Review Assessment: Checking Correctness Of Experiments:**

I assessed the sensibility of the experiments.

**Review Assessment: Thoroughness In Paper Reading:**

I read the paper thoroughly.

---

### Official Review · AnonReviewer2 · 2019-10-29
**Official Blind Review #2**

**Rating:** 3

**Review:**

This paper proposes to use autoencoder LSTM to learn car 3D crash simulation in an unsupervised manner.

Basically, it proposes to apply autoencoder LSTM, which is an existing and well-studied technique, to a very special task of car crash simulation. Neither a new method nor an in-depth investigation of the task itself is carried out. It is hard to identify contributions that are significant enough to support the publication in top conferences such as ICLR. Considering the fact that autoencoder LSTM is wildly employed in solving various unsupervised learning tasks, readers can hardly acquire knowledge in terms of novelty.

**Experience Assessment:**

I have published one or two papers in this area.

**Review Assessment: Checking Correctness Of Derivations And Theory:**

I carefully checked the derivations and theory.

**Review Assessment: Checking Correctness Of Experiments:**

I assessed the sensibility of the experiments.

**Review Assessment: Thoroughness In Paper Reading:**

I read the paper thoroughly.

---

### Author Response · Authors · 2019-11-13
**Down-stream application and contribution**

We see the main contributions of our paper for the ML audience as being able to easily handle geometric data by employing the recently introduced spectral basis, and introducing a time-based application for this. Although we apply it for a specific data set, the underlying approach can be used for other geometric time data sets, but haven't found suitable other data sources so far to apply this on. Yes, it is plugging together two known ideas, but the combination is new and strongly simplifies the handling of geometries in the NN context.

For the specific application of car crash, the aim is at this stage not to replace the numerical simulations, but to assist and simplify their handling and usage. For example, an application would be the early termination of the costly numerical simulation when the (quick to evaluate) LSTM-prediction shows an unwanted behavior, e.g. the wrong path of the bifurcation. Thereby saving hours of compute time and speeding up the design process. Compression of the simulation results is an application to further investigate, i.e. not saving all time steps since they can be reconstructed from a subset of time steps of a simulation. In particular, since in post-processing and visualization of the simulation the data does not need to be as precise as in the actual numerical simulation.

Adressing some of the specific questions
- Hard numbers on flops are not available, but rough statements from industry are overnight on 128-256 cores for a current size model. This statement is in some sense constant over the years, stronger compute cores usually result in larger FEM models.
- The underlying dynamics on the fine scale will be difficult to learn. The 'coarse' dynamic behavior is seemingly easier to learn, to some degree due to the employed spectral basis. Regarding that one, besides simplifying the treatment of geometries in a NN and giving a compact representation, the eigenvectors also capture geometric properties of the mesh to some extent.
- To consider relations between different parts would need quite a different approach, because the relations can be viewed as outer forces or boundary constraints. This would be relevant if one aims for replacing numerical simulations, but as outlined here, the aim at this stage is to simplify the usage of simulations.
- Furthermore, the joint treatment of the two beams/four parts is helpful for the prediction. In other words, learning a LSTM-model for the parts separately gives worse results.

---

### Decision · Program_Chairs · 2019-12-19

**Decision:**

Reject

**Comment:**


The paper proposes to train LSTMs to encode car crashes (a temporal sequence of 3D mesh representations).  Decoder LSTMs can then be used to 1) reconstruct the input or 2) predict the future sequence of structural geometry.  The authors propose to use a spectral feature representation based on prior work as input into the encoding LSTM.  The main contribution of the paper (based on the author response) is the introduction of this spectral feature representation to the ML community.  The authors used single 3D truck model to generate 205 simulations, of which 105 was used for training, and 100 for testing.  The authors presented reconstruction errors and TSNE visualization of the LSTM's reconstruction weights.

Discussion Summary:
The paper got three weak rejects.  The response provided by the authors failed to convince any of the reviewers to adjust their scores.  The authors did not provide a revision based on the reviewer comments.

Overall, the reviewers found the problem statement to be interesting.  However, they had concerns about the following:
1. It's unclear what is the main technical contribution of the work.
Several of the reviewers pointed out the lack of technical novelty.  From the writing, it's unclear if the proposed spectral feature representation is taken directly from prior work or there was some additional innovation in this submission.  Based on the author response, it seems the proposed feature representation is taken directly from prior work as the authors themselves acknowledge that the submission is taking two known ideas and combining them.  This can be made more explicit in the paper itself.

2. Lack of comparison with existing work and experimental analysis
There is no comparison against existing work on predicting 3D structure deformation over time. While the proposed representation is interesting, the is no comparison with other methods or other alternative representations.  Without any comparisons it is difficult to judge how the reconstruction error corresponding to actual reconstruction quality.  How much error is acceptable?  The submissions also fails to elucidate when the proposed representation should be used.  Is it better than alternative representations (use 3D mesh directly? use point clouds? use alternate basis functions?)

3. What is being learned by the model?
R3 pointed out that the authors mention that the model is trained in just half an hour and questioned whether the dynamics function is trivial to learn and that the only two parts of the 3D structure is analyzed.  The authors responded that the "coarse" dynamic is easier to learn than the "fine" scale dynamics.  Is what is learned by the model sufficient?  How well would a model that just modeled the car as a rigid object and predicted the position do?  The lack of comparison against baselines and alternative methods/representations makes it difficult to judge usefulness of the representation/approach that is presented.

4. The paper also has minor typos.
Page 5: "treat the for beams" --> "treat the four beams"
Page 7: "marrked" --> "marked"

Overall the paper addresses a interesting problem domain, and introduces a interesting representation to the ML community, but fails to do a proper experimental analysis showing how the representation compares to alternatives.  Since the paper does not claim the novelty of the representation as its contribution, it is essential that it performs a thorough investigation of the task and perform empirical studies comparing the proposed representation/method against baselines and alternatives.